# Effect of Short-Chain Fatty Acids on the Yield of 2,3-Butanediol by *Saccharomyces cerevisiae* W141: The Synergistic Effect of Acetic Acid and Dissolved Oxygen

Jiaxin Liu [†], Shanshan Sun [†], Yangcun Sun, Dean Liu, Jie Kang, Zeming Ye, Gang Song * and Jingping Ge *

Engineering Research Center of Agricultural Microbiology Technology, Ministry of Education & Heilongjiang Provincial Key Laboratory of Plant Genetic Engineering and Biological Fermentation Engineering for Cold Region & Key Laboratory of Microbiology, College of Heilongjiang Province & School of Life Sciences, Heilongjiang University, Harbin 150080, China

* Correspondence: songgang@hlju.edu.cn (G.S.); gejingping@126.com (J.G.); Tel.: +86-0451-86609106 (J.G.); Fax: +86-0451-86608046 (J.G.)

† These authors contributed equally to this work.

**Abstract:** As a platform chemical, 2,3-butanediol (2,3-BDO) has been widely used in various industrial fields. To improve the yield of 2,3-BDO produced by *Saccharomyces cerevisiae* W141, this paper explored the effects of exogenous short-chain fatty acids (SCFAs) as well as the synergistic effects of acetic acid and dissolved oxygen content on the yield of 2,3-BDO from the perspective of physiological metabolism. The results indicated that different SCFAs had different effects on the production of 2,3-BDO, and higher or lower concentrations of SCFAs were not conducive to the generation of 2,3-BDO. However, exogenically adding 1.0 g/L acetic acid significantly increased the yield of 2,3-BDO and the expression level of *bdh1*, a key gene in the synthesis of 2,3-BDO ($p < 0.05$). In addition, a dissolved oxygen concentration of 4.52 mg/L was proven to be the optimal condition for 2,3-BDO production. When the dissolved oxygen content and acetic acid concentration were 4.52 mg/L and 1.0 g/L, respectively, the maximum yield of 2,3-BDO was 3.25 ± 0.03 g/L, which was 66.59% higher than that produced by *S. cerevisiae* W141 alone. These results provide methodological guidance for the industrial production of 2,3-BDO by *S. cerevisiae*.

**Keywords:** 2,3-butanediol; *Saccharomyces cerevisiae*; short-chain fatty acids; dissolved oxygen

## 1. Introduction

2,3-Butanediol (2,3-BDO) is a natural metabolite and important potential platform chemical that can be used as an important precursor material for the synthesis of chiral reagents and ligands and has extremely high industrial value, specifically in the solvent, fuel, polymer, cosmetics and medicine fields [1–4]. Among the bacterial strains that can produce 2,3-BDO, *Klebsiella* sp. (e.g., *Klebsiella oxytoca*) and *Enterobacter* sp. (e.g., *Enterobacter cloacae*) are mostly opportunistic pathogens and are not suitable for green and safe industrial production [5]. In addition, little information is known about the single steric configuration of 2,3-BDO produced by bacteria, which results in greater costs for industrial separation and purification and a relatively tedious preparation process [6]. Remarkably, *Saccharomyces cerevisiae* is not only recognized as a GRAS (generally regarded as safe) microorganism but can also produce the most extensively applied configuration of 2,3-BDO, i.e., meso-2,3-BDO [7–9]. However, the low yield of 2,3-BDO produced by *S. cerevisiae* has become a bottleneck problem limiting large-scale industrial production. Therefore, there is an urgent need to develop an efficient and stable method for producing 2,3-BDO from *S. cerevisiae*.

To date, several strategies have been used to improve the yield of 2,3-BDO by *S. cerevisiae*, for example, reducing the formation of by-products through the gene knock-out method [10–12], enhancing the glycolysis pathway [13] and exogenously expressing 2,3-BDO synthesis pathway genes [14,15]. For these strategies, eliminating the yields of ethanol

and glycerol and rebalancing cellular redox were the key factors for increasing the yield of 2,3-BDO [10,16]. In previous studies, our research group studied the above methods to improve 2,3-BDO yields with *S. cerevisiae* W141, but the results were not significant [4,12–14], indicating that other limiting factors might inhibit 2,3-BDO production.

In 1995, Garg et al. found that acidic conditions increased diol synthesis (3–7-fold), while alkaline conditions favored the formation of organic acids, resulting in a decrease in 2,3-BDO production [6]. Xu et al. confirmed that when the acetic acid concentration reached a certain level, pyruvate was increasingly utilized in a 2,3-BDO-generating pathway [17], and intracellular acidification could be avoided through the transformation of metabolism from acidic products to neutral products [18]. Short-chain fatty acids (SCFAs) have received much attention in recent years. Although they do not directly participate in the formation of 2,3-BDO, the addition of these SCFAs increases the yield of 2,3-BDO, specifically in the production of 2,3-BDO by *Eenterobacter aerogenes*, *Klebsiella pneumoniae*, and *Paenibacillus polymyxa* [19–22].

*S. cerevisiae* commonly undergoes acid stress in industrial applications, which greatly affects the yields of 2,3-BDO and other metabolites [23–25]. Some reports proved that ethanol fermentation by *S. cerevisiae* was disturbed under acid stress [25,26], while we also found that the addition of 1.5 g/L acetic acid could increase the 2,3-BDO yield of *S. cerevisiae* W141, and acetic acid potentially acted as a signaling molecule in the quorum sensing system to increase 2,3-BDO production by *S. cerevisiae* [4]. However, there are few studies on the effects of other SCFAs on the production of 2,3-BDO in *S. cerevisiae*.

In this study, we hypothesized that the addition of SCFAs (formic acid, acetic acid, propionic acid, butyric acid and valeric acid) would improve the production of 2,3-BDO by *S. cerevisiae* W141, as indicated by enzyme activities, mRNA expression levels and fermentation abilities. The purpose of this study was to (1) clarify the optimal fermentation conditions for 2,3-BDO production by *S. cerevisiae* W141 through the exogenous addition of five different concentrations of SCFAs and (2) investigate whether the exogenous addition of SCFAs could affect the expression of key genes for the synthesis of 2,3-BDO. These outcomes will help to increase the yield of 2,3-BDO produced by *S. cerevisiae* and provide new ideas for the industrial production of 2,3-BDO.

## 2. Material and Methods

### 2.1. Strains, Media and Cultivation Conditions

*S. cerevisiae* W141 (CICC 31526), a diploid wild-type strain, was isolated from soil in Heilongjiang Province, China, and preserved at the Key Laboratory of Microbiology, Heilongjiang University, China. *S. cerevisiae* cells were grown at 30 °C in yeast-peptone-dextrose-adenine (YPD) medium (10.0 g/L yeast extract, 20.0 g/L Bacto peptone and 20.0 g/L glucose) to harvest seed cultures. For fermentation, seed cultures were inoculated at 5% inoculum (*v/v*) into glucose fermentation medium (80.0 g/L glucose, 20.0 g/L Bacto peptone, 3.4 g/L yeast nitrogen base, 10.0 g/L $(NH_4)_2SO_4$, 11.8 g/L $KH_2PO_4$, and 3.0 g/L $K_2HPO_4$). Fermentation was carried out at 30 °C with a shaking speed of 140 r/min in 200 mL/500 mL Erlenmeyer flasks with a rubber stopper for 120 h to assess the 2,3-BDO yield. All chemical reagents used in the present study were obtained from Tianjin Kemiou Chemical Reagent Co., Ltd., (Tianjin, China).

### 2.2. Effect of SCFAs on the Fermentation Performance of S. cerevisiae W141

First, 0.5, 1.0, 1.5, 2.0 and 2.5 g/L of SCFAs (formic acid, acetic acid, propionic acid, butyric acid and valeric acid) were added to the glucose fermentation medium of *S. cerevisiae* W141, and 8 g/L of acetoin was used to improve the fermentation effect. The fermentation medium without SCFAs was used as a control. Fermentation was carried out at 30 °C with a shaking speed of 140 r/min in flasks for 120 h. The pH value, $OD_{600 \text{ nm}}$, substrate concentration, and 2,3-BDO content were measured from samples collected at intervals of 12 h. All the experimental group samplings were performed in triplicate.

To investigate whether the effect of the exogenous addition of SCFAs on the yield of 2,3-BDO was facilitated by $H^+$ or acid ions, *S. cerevisiae* W141 was cultured in media with different initial pH values (3, 4, 5, 6, 7 and 8) adjusted with inorganic acid ($H_2SO_4$) and shaken at 140 r/min for 120 h at 30 °C. Additionally, the pH value, $OD_{600\,nm}$, and 2,3-BDO content were measured from samples collected at intervals of 12 h. All the experimental group samplings were performed in triplicate.

### 2.3. Detection of mRNA Expression by qRT-PCR

The *S. cerevisiae* W141 strain with the exogenous addition of five different SCFAs was cultured at 30 °C and 140 r/min. RNA was extracted from *S. cerevisiae* W141 with the HiPure Yeast RNA Kit (#R4182-02, Magen Biotech Co. Ltd., Shanghai, China). The RNA concentration and quality were determined using a Nanodrop 2000 spectrophotometer (Thermo Scientific Co., Ltd., Waltham, MA, USA) by measuring the absorbance ratio at 260/280 nm (A260/280 ratio). Reverse transcription was performed using LunaScript® RT SuperMix (New England Biolabs, Inc., Paris, France) following the operation manual. qRT-PCR was performed in triplicate with a PrimeScript™ RT Reagent Kit (Takara Bio Inc., Shiga, Japan) using SYBR Green-based detection with a 7500 Real-Time PCR System (Applied Biosystems, Inc., Waltham, MA, USA). Primers for qRT-PCR were designed using Primer5 software, and *bdh1*, *hog1p*, and *haa1p* were expressed with the following primers: *bdh1* 5′- TTTGCTGAACAAGTCGTAGTC, 3′- CCCAGTTTCTTGGCCATTTC; *hog1p*, 5′- ACTACTGGTGCCCAAACTAAT, 3′- TGTTCCCATAAACTCGGCTAAA; and *haa1p*, 5′- GACATCTAAGAAAGTCCCCT, 3′- CTGTCTAAAAATGTGCTCGT. *bdh1* is the gene encoding butanediol dehydrogenase, and *hog1p* and *haa1p* are associated with acid tolerance [27] (60 h sampling). The housekeeping gene 18S rRNA was chosen to normalize the RNA amounts as an internal control. The CT values of each sample were analyzed using the relative quantitative $2^{-\Delta\Delta CT}$ method and were corrected using the housekeeping gene 18S rRNA [28].

### 2.4. Determination of Key Enzyme Activities

The strain *S. cerevisiae* W141 was inoculated into YPD liquid medium overnight, transferred to 100 mL/250 mL fermentation medium with an initial glucose concentration of 80.0 g/L, and cultured at 200 r/min at 30 °C. Samples were collected every 6 hours and centrifuged at room temperature at 13,000 r/min for 8 min to harvest the cell precipitate. Then, 400 μL of extracting solution (provided by the kit) was added for every 2 million bacteria or cells. The cell mixture was degraded by ultrasound (20% power for 3 s, 10 s intervals, repeated 30 times) and centrifuged at $8000\times g$ for 10 min (4 °C). The supernatant was taken and placed on ice for enzyme activity measurement.

The Micro Pyruvate Decarboxylase (PDC) Assay Kit and Micro Acetolactate Synthase (ILV2) Assay Kit were purchased from Solarbio Science & Technology Co., Ltd., Beijing, China. A 2,3-butanediol dehydrogenase (BDH1) ELISA kit was purchased from Mlbio Bioengineering Co., Ltd., Shanghai, China. Enzymatic assays for the three enzymes were performed in accordance with the protocol provided by the manufacturer.

### 2.5. Optimization of Fermentation Conditions

*S. cerevisiae* are facultative anaerobic organisms that require oxygen during the fermentation process, and different oxygen concentrations have various effects on the products. In this study, to acquire an optimal oxygen supply, the dissolved oxygen content was controlled by changing the number of gauze layers (4, 8 and 12) during fermentation, the rotation speed of the shaker (100, 140 and 180 r/min) and the working volume of 500 mL flasks (100, 150 and 200 mL) using a portable dissolved oxygen detector (Mettler Toledo S4-Standard kit, Zurich, Switzerland) for dissolved oxygen detection and simultaneously adding 1.0 g/L acetic acid. A standard orthogonal optimization experiment $L_9(3^3)$ was performed, and the yield of 2,3-BDO was set as an objective function. The effects of the three process parameters of the gauze layer number, shaking speed and working volume/500 mL

on the yield of 2,3-BDO were studied, and each process parameter had three different levels (Table 1).

**Table 1.** Orthogonal experimental design $L_9(3^3)$ for the three process parameters and three different levels.

| Process Parameters | Level 1 | Level 2 | Level 3 |
|---|---|---|---|
| A: gauze layer | 4 | 8 | 12 |
| B: shaker speed (r/min) | 100 | 140 | 180 |
| C: work volume (mL) | 100 | 150 | 200 |

The effects on the 2,3-BDO yield were determined according to each experimental design, and the order of the effects of the three parameters on the 2,3-BDO concentration was selected based on the R value. The optimal combination of fermentation conditions was determined on the basis of K values [29].

$$KiA = \Sigma \ 2,3\text{-BD yield at } Ai.$$

$$RA = max\{KiA\}\text{-}min\{KiA\}.$$

*2.6. Analysis of 2,3-BDO Yield*

High-performance liquid chromatography (HPLC, Shimadzu LC-10ATvp, Shanghai, China) with an HPX-87H column (300 mm × 67.8 mm, Aminex HPX-87H ion exclusion column) was used to detect 2,3-BDO yield at 65 °C with a refractive index detector (RID–10 A). Moreover, 0.005 M $H_2SO_4$ was used as the elution buffer with a flow rate of 0.8 mL/min, and the analysis time was 20 min. The temperature of the differential detector was 40 °C, and the temperature of the chromatographic column was 156 °C, which was controlled at 65 °C [11].

*2.7. Statistical Analysis*

Statistical analysis of the data was performed using Statistical Product and Service Solutions (SPSS), with the results presented as X ± SD. Probability values of less than 0.05 ($p < 0.05$) were considered to indicate a significant difference. Origin 2022b was used for the statistical analysis of the data and to generate charts.

**3. Results**

*3.1. Effect of SCFAs on the Fermentation Performance of S. cerevisiae W141*

The results of 25 sets of experiments (Figure 1A) showed that the addition of formic acid, butyric acid and valeric acid had a severe inhibitory effect on the production of 2,3-BDO by *S. cerevisiae* W141, particularly at higher concentrations. Among these SCFAs, formic acid had the most significant inhibitory effect ($p < 0.05$). When 0.5 g/L formic acid was added, the maximum 2,3-BDO production of *S. cerevisiae* W141 was 1.25 ± 0.04 g/L after 60 h fermentation, which was significantly lower (by 35.90%) than that of the control (1.95 ± 0.06 g/L for 60 h) ($p < 0.05$).

As the concentration of formic acid increased, the ability of *S. cerevisiae* W141 to synthesize 2,3-BDO was gradually reduced and was completely inhibited when the formic acid concentration reached 2.5 g/L. Valeric acid had a similar inhibition trend to that of formic acid, which inhibited the synthesis of 2,3-BDO by *S. cerevisiae* W141 more strongly than butyric acid (Figure 1A), particularly at 2.0 and 2.5 g/L. Moreover, trace amounts of 2,3-BDO were still produced in the propionic acid treatment, and this phenomenon was not observed in the other four acid treatments at 2.5 g/L. Furthermore, 2,3-BDO production reached a maximum value (2.19 ± 0.07 g/L, 60 h) when 0.5 g/L propionic acid was added, which was increased by 12.31% compared with that of the control group, and higher concentrations did not completely inhibit the synthesis of 2,3-BDO as when formic acid, butyric acid, acetic acid and valeric acid were added.

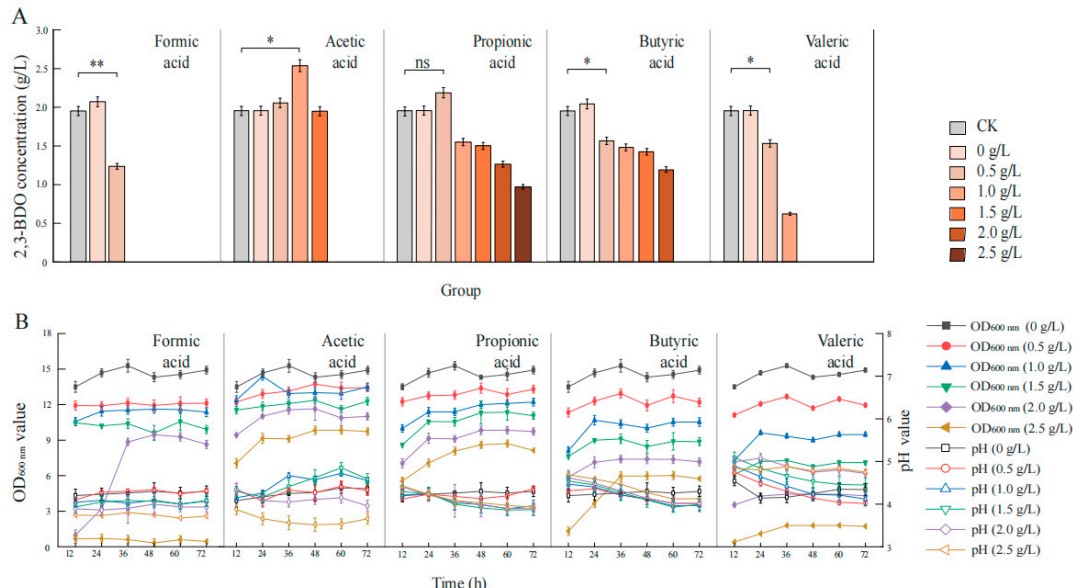

**Figure 1.** Fermentation results obtained by the exogenous addition of five types of short-chain fatty acids. (**A**) Bar graph showed the effect of short-chain saturated fatty acids at different concentrations on the yield of 2,3-BDO from *S. cerevisiae* W141. The vertical coordinates showed the 2,3-BDO yields. (**B**) The point line graph showed the effects of five different concentrations of short-chain fatty acids on pH value and the growth of *S. cerevisiae* W141. Solid points indicated changes in $OD_{600\,nm}$ and hollow points indicated changes in pH. Significance levels (* $p < 0.05$; ** $p < 0.01$; ns: no significance).

Remarkably, the addition of 1.0 g/L acetic acid was considered to be the optimal concentration for increasing the yield of 2,3-BDO. As the concentration increased, the 2,3-BDO production of *S. cerevisiae* W141 reached a maximum value after 60 h ($2.53 \pm 0.08$ g/L), a significant increase of 29.74% compared with that of the control group ($p < 0.05$), and the 2,3-BDO production was also higher than that of the other groups. However, in contrast to that of propionic acid, the synthesis of 2,3-BDO was abruptly stopped when the concentration of acetic acid was greater than or equal to 2.0 g/L. These results showed that formic acid, butyric acid and valeric acid inhibited the fermentation performance of *S. cerevisiae* W141, while formic acid had the strongest inhibitory effect. However, lower concentrations of propionic and acetic acids promoted 2,3-BDO synthesis by *S. cerevisiae* W141.

When acetic acid was added exogenously, the pH value fluctuated more than those of other groups, and the overall pH value remained between 3.5 and 5 (Figure 1B). The acidic environment was suitable for the production of 2,3-BDO. When 0.5 and 1.0 g/L acetic acid were added, the growth of the strain was not significantly different from that in the control group, indicating that 0.5 and 1.0 g/L acetic acid could not inhibit the growth of *S. cerevisiae* W141, while formic acid, propionic acid, butyric acid and valeric acid had a great effect on the growth of the strain (Figure 1B).

A high concentration of SCFAs had the lowest correlation with 2,3-BDO production, while a low concentration of SCFAs had a low correlation with 2,3-BDO production. Moreover, both high and low polyhydric alcohol concentrations were not conducive to the generation of 2,3-BDO (Figure 2).

Different initial fermentation pH values of *S. cerevisiae* W141 were adjusted using inorganic acid ($H_2SO_4$). The results showed that the growth of *S. cerevisiae* W141 and synthesis of 2,3-BDO were inhibited when the initial pH was neutral or alkaline (Figure 3A), and $H^+$ had little effect on the growth and the synthesis of 2,3-BDO within a specific pH range (pH = 3–5) (Figure 3B).

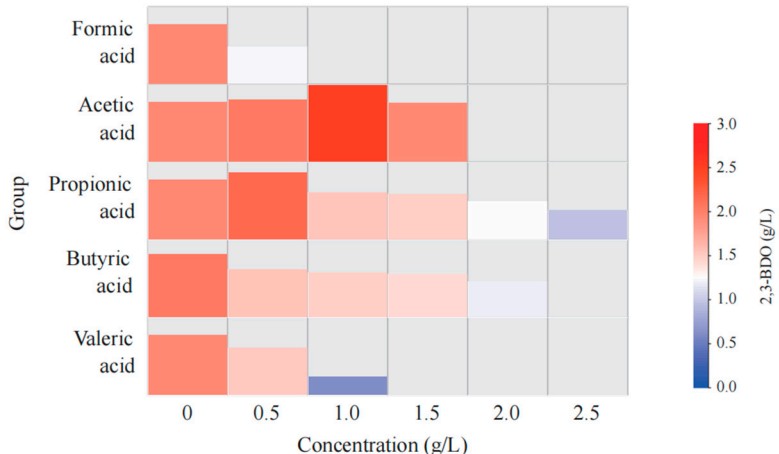

**Figure 2.** The changes in the 2,3-BDO yield by *S. cerevisiae* W141 at 60 h.

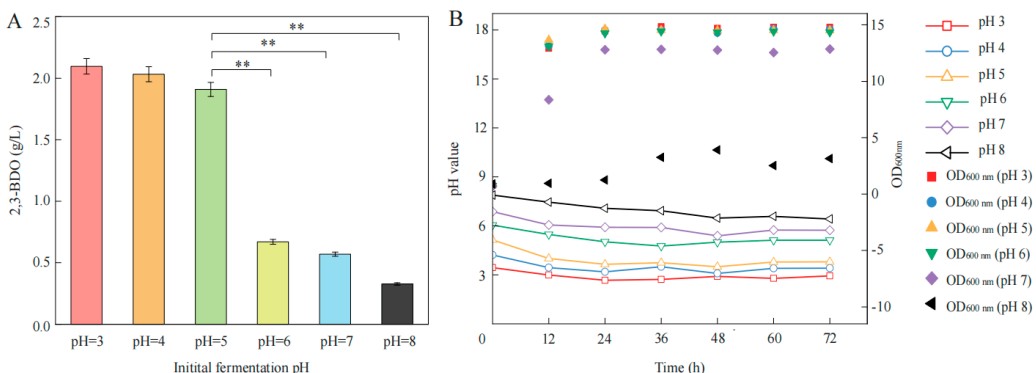

**Figure 3.** The effect of pH adjusted by inorganic acid (H$_2$SO$_4$) on the (**A**) 2,3-BDO production by *S. cerevisiae* W141, and (**B**) on the OD$_{600 \text{ nm}}$ values of *S. cerevisiae* W141 with time. Significance levels (** $p < 0.01$).

The fermentation results of the five SCFAs with optimal concentrations (formic, propionic, butyric and valeric acid: 0.5 g/L and acetic acid: 1.0 g/L) showed that pH had a greater effect on the yield of 2,3-BDO in the range of 4.04–4.72. These results demonstrated that different SCFA ions play a crucial role in the synthesis of 2,3-BDO (Figures 2 and 4).

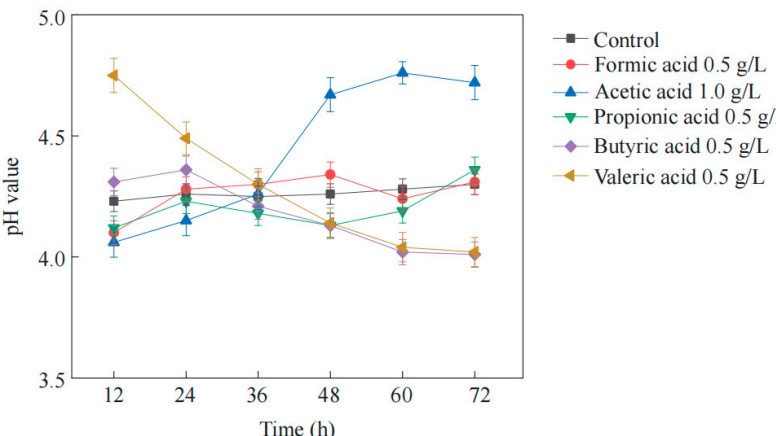

**Figure 4.** The dynamic changes in pH at optimal concentrations of five short-chain fatty acids.

### 3.2. Effect of SCFAs on the mRNA Expression and Related Enzyme Activities of S. cerevisiae W141

Similar to the above results, different SCFAs had different effects on the expression of *bdh1* (Figure 5). Since the exogenous addition of acetic acid could increase the yield of 2,3-BDO, 1.0 g/L acetic acid was used as an example to reveal the mechanism by which acetic acid promoted the increase in 2,3-BDO production. The results showed that after 1.0 g/L acetic acid was added exogenously, the relative expression level of *bdh1* increased by 1.55-fold ($p < 0.05$) (Figure 6A), indicating that the addition of acetic acid could improve the gene expression level of the 2,3-BDO metabolic pathway, thus further improving the yield of 2,3-BDO. Meanwhile, the relative expression levels of *haa1p* and *hog1p* were also increased by 1.45- and 1.37-fold, respectively, and the differences were significant ($p < 0.05$) (Figure 6B), suggesting that the addition of acetic acid caused a response to acid stress by the organism.

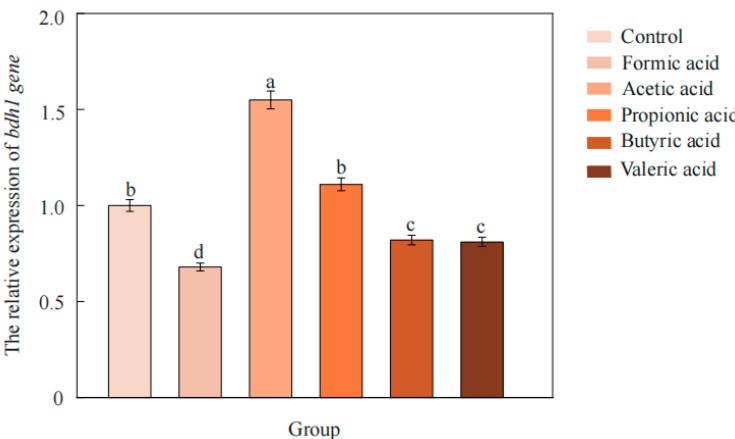

**Figure 5.** The changes in *bhd1* gene expression in *S. cerevisiae* W141 under optimal concentrations of five short-chain fatty acids. a, b, c and d are the results obtained by SPSS significance analysis.

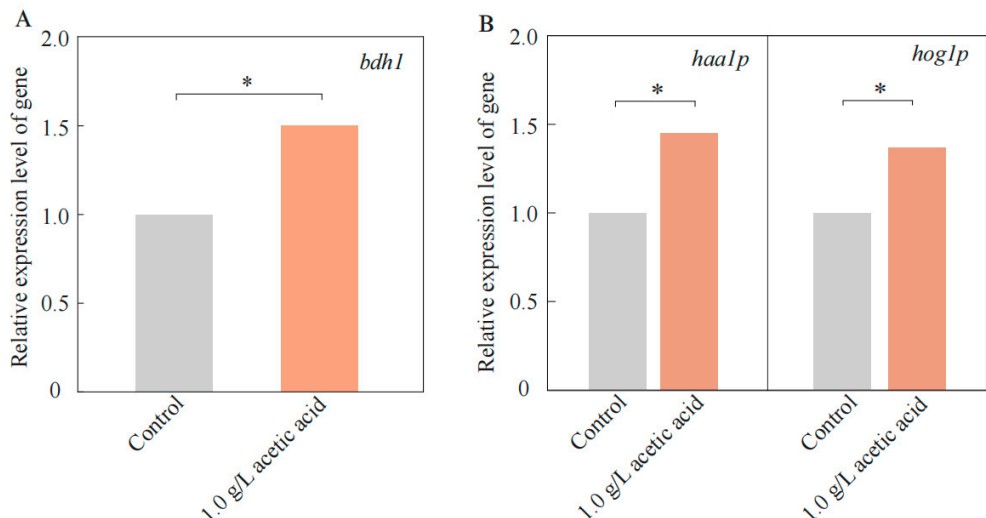

**Figure 6.** The effect of adding 1 g/L acetic acid on the gene expressions of *bdh1* (**A**), *haa1p* and *hog1p* (**B**) of *S. cerevisiae* W141. Significance levels (* $p < 0.05$).

Considering the changes in the key enzyme activities of 2,3-BDO metabolism, it was found that the exogenous addition of 1.0 g/L acetic acid could increase the activity of PDC, ILV2 and BDH1 enzymes by 28.87%, 29.41% and 32.26%, respectively ($p < 0.05$). These results indicated that the exogenous addition of 1.0 g/L acetic acid improved the activity of PDC and the key enzymes of 2,3-BDO synthesis in the central metabolic pathway of *S. cerevisiae* W141 (Table 2).

**Table 2.** Results of enzyme activity of exogenously added 1 g/L of acetic acid in *S. cerevisiae* W141.

| Enzyme Activity | *S. cerevisiae* W141 | *S. cerevisiae* W141+1.0 g/L Acetic Acid | Rate |
|---|---|---|---|
| PDC (U/mg) | 0.494 ± 0.01b | 0.685 ± 0.02a | Increase 28.87% |
| ILV2 (U/mg) | 0.048 ± 0.001b | 0.068 ± 0.004a | Increase 29.41% |
| BDH1 (U/mg) | 0.042 ± 0.001b | 0.062 ± 0.001a | Increase 32.26% |

Note: a and b, are the results obtained by SPSS significance analysis.

### 3.3. Fermentation of 2,3-BDO with Optimal Dissolved Oxygen Content

An optimal dissolved oxygen content is a prerequisite for achieving increased 2,3-BDO production. Under laboratory conditions, the dissolved oxygen content can be effectively controlled by the number of gauze layers, the shaker speed and the working volume in the flasks during the fermentation process with an orthogonal test. From the single factor results, it was found that all three factors could affect 2,3-BDO production (Figure 7). Factor analysis of the orthogonal fermentation test showed that the optimal fermentation conditions were 8 gauze layers and a 200 mL/500 mL working volume with 100 r/min (A2B1C3) (Tables 3 and 4), under which the 2,3-BDO yield was 2.56 g/L and dissolved oxygen content was 4.52 mg/L, 31.28% higher than that of the control, and the expression of the *bdh1* gene increased 1.31-fold ($p < 0.05$) (Table S1). Under the optimal dissolved oxygen conditions, simultaneously adding 1.0 g/L acetic acid could increase the yield of 2,3-BDO by 26.95% (3.25 ± 0.03 g/L) compared with that of the control group and by 66.59% compared with that of the group without optimized fermentation conditions.

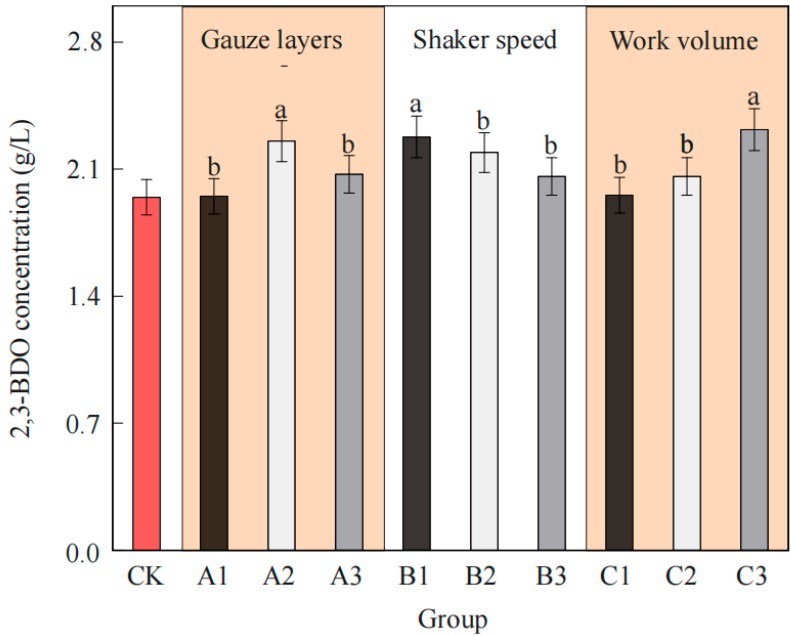

**Figure 7.** Fermentation results obtained by changing culture conditions. a and b, are the results obtained by SPSS significance analysis.

**Table 3.** Orthogonal fermentation test 2,3-butanediol production.

| Serial Number | Experiment Table | 2,3-BD Yield (g/L) | Dissolved Oxygen (mg/L) |
|---|---|---|---|
| 1 | A1B1C1 | 2.34 ± 0.21 [a]c | 5.61 |
| 2 | A1B2C2 | 2.25 ± 0.11 d | 6.57 |
| 3 | A1B3C3 | 2.08 ± 0.10 h | 7.22 |
| 4 | A2B1C3 | 2.56 ± 0.16 a | 4.52 |
| 5 | A2B2C1 | 2.37 ± 0.12 b | 5.53 |
| 6 | A2B3C2 | 2.17 ± 0.12 f | 6.89 |
| 7 | A3B1C2 | 2.18 ± 0.14 f | 4.13 |
| 8 | A3B2C3 | 2.22 ± 0.10 e | 6.37 |
| 9 | A3B3C1 | 2.12 ± 0.16 g | 6.94 |

Note: The table columns include multiple comparisons. [a] Mean of three determinations ± standard deviation, and a–h are the results obtained by SPSS significance analysis. A, B and C indicate the gauze layers, shaker speed and working volume, respectively.

**Table 4.** Analysis of orthogonal fermentation test factors.

| | | Level A | Level B | Level C |
|---|---|---|---|---|
| | K1 | 6.67 | 7.08 | 6.83 |
| Level sum | K2 | 7.10 | 6.84 | 6.60 |
| | K3 | 6.52 | 6.37 | 6.86 |
| | K1 | 2.22 | 2.36 | 2.28 |
| Level average value | K2 | 2.37 | 2.28 | 2.20 |
| | K3 | 2.17 | 2.12 | 2.29 |
| Range | | 0.20 | 0.24 | 0.09 |
| Optimal Factor | | A2 | B1 | C3 |

Note: A, B and C indicate the gauze layers, working volume, and shaker speed, respectively.

## 4. Discussion

As a platform chemical, 2,3-BDO has been widely used in various fields, such as food and medicine [10]. The microbial production of 2,3-BDO for industrial applications involving bacteria and yeast in isolation and metabolic engineering for the efficient production of 2,3-BDO has been reported [20]. In this study, we evaluated the effect of the exogenous addition of different types and concentrations of acids and dissolved oxygen contents on the fermentation performance of *S. cerevisiae* W141, which resulted in an increase in 2,3-BDO production by *S. cerevisiae* W141.

It has been shown that acid can promote the production of 2,3-BDO by bacteria. For fermentation by *P. polymyxa*, the addition of acetic acid, propionic acid and pyruvic acid was shown to increase the production of 2,3-BDO, while 1.0 g/L acetic acid could increase the production of 2,3-BDO by 2-3-fold [19]. The addition of a certain amount of acetic acid to the culture of *E. aerogenes* ATCC 29,007 resulted in a significant increase in 2,3-BDO production [20]. Other reports also showed that the yield of 2,3-BDO increased significantly under acid fermentation conditions [30,31]. These results were in close agreement with the findings of this study. Therefore, it is further demonstrated that moderate acidity can promote the production of 2,3-BDO. In the present study, 2,3-BDO production by *S. cerevisiae* W141 decreased after the addition of formic, butyric and valeric acids, and its growth was significantly inhibited (Figure 1). This phenomenon may be due to the functional limit of the tricarboxylic acid cycle (TCA) of *S. cerevisiae*, resulting in an inability to produce enough energy for the synthesis of short- and medium-chain fatty acids (MCFAs, C4:0-C12:0) [32].

In addition, after adding formic acid, the growth of *S. cerevisiae* W141 was significantly inhibited, and the production of 2,3-BDO was reduced. The transcriptome results showed that when formic acid is present in the fermentation environment, it accelerates energy consumption by the cell to activate biotransformation, the stress response and transmembrane transport to resist formic acid [33]. However, formic acid can inhibit mitochondrial

cytochrome oxidase at the end of the respiratory chain, and low concentrations of formic acid can cause a rapid burst of reactive oxygen species in the mitochondria that can kill yeast cells [34,35].

Therefore, adding formic, butyric or valeric acids is not conducive to the growth of yeast cells and is even more unfavorable for the synthesis of 2,3-BDO or other metabolites. Interestingly, different results were obtained after adding acetic and propionic acids, which were beneficial for 2,3-BDO synthesis by *S. cerevisiae* W141 at specific concentrations. Propionic acid had a relatively small stimulatory effect on the ability of *S. cerevisiae* W141 to produce 2,3-BDO. A low concentration of propionic acid (0.5 g/L) increased 2,3-BDO production by 12.31%. Furthermore, when the propionic acid concentration was 2.5 g/L, *S. cerevisiae* W141 could still produce 2,3-BDO, which was distinctly different from the other four acids.

In the presence of 2.0 g/L acetic acid, the biosynthesis of 2,3-BDO by *S. cerevisiae* W141 ceased because acetic acid, similar to pyruvate, was converted to acetyl-CoA in the TCA cycle by *S. cerevisiae* mitochondria, and a higher concentration of acetic acid led to an imbalance in intracellular acetylation, which was detrimental to cell survival [36]. Hence, the 2,3-BDO yield of *S. cerevisiae* W141 increased after the addition of an appropriate amount of acetic acid. When 1.0 g/L acetic acid was added, the relative gene expression of *bdh1*, the activity of BDH1 and the 2,3-BDO production increased by 1.55-fold, 32.26% and 29.74%, respectively, compared with the control group. We observed that *S. cerevisiae* W141 enhanced tolerance to acetic acid through a strategy of upregulating the expression of *haa1p* and *hog1p*, key genes for this organism's response to acid stress, thereby enhancing acetic acid tolerance. The transcription factor encoded by *haa1p* regulates most of the genes related to acetic acid tolerance [37], such as the acid-tolerance gene *hog1p*. Hog1p phosphorylates the channel protein Fps1p, causing it to undergo endocytosis and reducing the entry of acetic acid into the cell [38]. Furthermore, the results of this study through the exogenous addition of inorganic acid ($H_2SO_4$) showed that the effect of the exogenous addition of five different SCFAs on the yield of *S. cerevisiae* W141 2,3-BDO was facilitated by acid ions rather than $H^+$.

In this study, we also evaluated the effect of dissolved oxygen content on 2,3-BDO yield, and different dissolved oxygen contents resulted in significant changes in the 2,3-BDO yield of *S. cerevisiae* W141. Dissolved oxygen plays a crucial role in the production and metabolism of industrial fermentation microorganisms. Oxygen utilization is a key factor for glucose catabolism in *S. cerevisiae*, and regulating the oxygen supply threshold is key to the regulation of *S. cerevisiae* metabolic pathways. If the dissolved oxygen content is decreased, *S. cerevisiae* metabolism shifts to produce glycerol and other metabolic by-products. In contrast, when the dissolved oxygen content is increased, although the maintenance of cell viability and the formation of *S. cerevisiae* biomass is promoted, these conditions also reduce ethanol biosynthesis [39]. Hence, when the amount of dissolved oxygen in the fermentation environment gradually decreased, the ethanol metabolic capacity of *S. cerevisiae* W141 gradually decreased, and the production of 2,3-BDO and glycerol first increased and then decreased. When the dissolved oxygen content was 4.52 mg/L, the largest yield of 2,3-BDO was obtained, reaching 2.56 ± 0.13 g/L. These results were similar to results reported for the optimal amount of dissolved oxygen for metabolically engineered *Klebsiella oxytoca* KMS005 to synthesize 2,3-BDO, in which the optimal amount of dissolved oxygen for meso-2,3-BDO production by *Bacillus subtilis* was lower than that of *S. cerevisiae* W141 [40,41]. These results showed that different microorganisms have their own optimal dissolved oxygen conditions when producing 2,3-BDO. NADH is consumed intracellularly and requires the *noxE* gene as well as the presence of intracellular dissolved oxygen [42]. When the dissolved oxygen content was 4.52 mg/L, the oxygen input was strictly controlled, and the consumption of intracellular NADH by the *noxE* gene was inhibited, thus resulting in a 36.36% increase in the production of 2,3-BDO by *S. cerevisiae* W141-E (*noxE*-overexpressing strain) compared with that of the control group. Moreover, the addition of 1.0 g/L acetic acid increased the 2,3-BDO yield of *S. cerevisiae* W141-E by

20.31% compared with that of the control group. This further confirmed that acid stress and dissolved oxygen levels may regulate 2,3-BDO biosynthesis by *S. cerevisiae* W141 by altering the intracellular redox balance. In this study, the fermentation conditions for the production of 2,3-BDO by *S. cerevisiae* W141 were optimized by adding different kinds and concentrations of SCFAs and controlling the content of dissolved oxygen, which will assist in improving the production and productivity of 2,3-BDO industrial fermentation.

## 5. Conclusions

The production of 2,3-BDO by *S. cerevisiae* was significantly improved by adopting certain strategies. In this study, at a dissolved oxygen level of 4.52 mg/L and with the exogenous addition of 1.0 g/L acetic acid, the production of 2,3-BDO by *S. cerevisiae* W141 reached a maximum compared with that of the control group. The results of the exogenous addition of inorganic acid ($H_2SO_4$) showed that the SCFA ions influenced the production of 2,3-BDO. It was also clarified that the exogenous addition of a moderate amount of acetic acid increased the production of 2,3-BDO by increasing the gene expression of *bdh1*. This study greatly enhances our understanding of the microbial production of 2,3-BDO and provides a solid theoretical basis for further improving the yield of 2,3-BDO in industrial production.

**Supplementary Materials:** The following supporting information can be downloaded at: https://www.mdpi.com/article/10.3390/fermentation9030236/s1, Table S1: Relative expression of bdh1 gene in *S. cerevisiae* W141 at 4.52 mg/L dissolved oxygen level.

**Author Contributions:** J.L. and S.S.: Methodology, Data curation, Writing—original draft, Writing—review and editing. Y.S.: Formal analysis. D.L.: Formal analysis. J.K.: Investigation. Z.Y.: Validation. G.S.: Resources, Supervision. J.G.: Resources, Supervision and Funding. All authors have read and agreed to the published version of the manuscript.

**Funding:** This work was supported by the Key Program of Heilongjiang Provincial Natural Science Foundation of China [No. ZD2020C008]; the Scientific Research Project of Ecological Environment Protection of Heilongjiang Provincial Department of Ecological Environment (HST2022T004); and the Heilongjiang University Graduate Innovation Research Project (YJSCX2022-091HLJU).

**Institutional Review Board Statement:** Not applicable.

**Informed Consent Statement:** Not applicable.

**Data Availability Statement:** The data presented in this study are available on request from the corresponding author.

**Conflicts of Interest:** The authors declare no conflict of interest.

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
