# Peer review of "Effect of Short-Chain Fatty Acids on the Yield of 2,3-Butanediol by Saccharomyces cerevisiae W141: The Synergistic Effect of Acetic Acid and Dissolved Oxygen"

_fermentation, doi:10.3390/fermentation9030236_

Round 1

Reviewer 1 Report

The undertaken research was supported by five funds, the national equivalents of the NSF, DOE, and USDA in the USA and a few others, the most impressive for me was the fund supporting the “Innovation of Modern Agricultural Industry Technology”. Surprisingly, the research per se was very naïve and modest. First, the authors strongly believe that all bacteria are opportunistic pathogens (including all species of genera Klebsiella and Enterobacter, although many of them are not only benign to humans but also produce a lot of good products and play an important beneficial role in soils and native waters). Anyway, they switched from active producers of the target product, 2,3-Butanediol (up to 40-60 g/L in batch fermentation) to the baker yeasts producing just 2-3 g/L!

Second, to optimize the poor performance of Saccharomyces cerevisiae, they tested three levels of aeration (it reminded me of a standard wet lab for undergrad students in biotech colleges 30-50 years ago) and the effects of added fatty acids homologs from C1 (formic acid) to C5 (valeric acid). Both effects turned out to be well within experimental scatter apart from clearly expressed inhibition at high concentrations of acids. However, I do not see any significant improvement in the biotech performance of the yeasts.

Frankly speaking, I do not see any messages in this MS useful for the journal ‘Fermentation’ reader. There are no significant bioindustry-related results or any fundamental novelty. The experiments are simplistic. English is horrible (if Editor will accept the MS, it should be fundamentally revised, and 90% of the prose should be replaced to reach the status of professional quality).

I should also mention that the authors added a completely redundant molecular part: qPCR for detecting three mRNA and measuring some enzymatic activity. Besides, the Discussion is oversaturated by extracts from biochemistry and molecular biology textbooks. It is exclusively cosmetic polish, the discussion matters exceed the experimental scope, and qPCR did not carry any useful information.

I also found evidence of inadequate training of the authors in the fermentation technique:

Lines 139-140, they used the Mettler Toledo oxygen probes to record dissolved oxygen concentration in regular shaking flasks, which is impossible, such probes can work ONLY in bioreactors!

Line 95, they discuss the effect of “H+ or acid ions”, C- in freshman chemistry

Line 20 (abstract) and below, DO concentration is expressed as vvm, the air volume per culture volume per min. Such units are used for the old-style mass-transfer characterization, NOT concentration!

Line 19, gene bdh1 was called ‘a key gene in the synthesis of 2,3-BDO’, while it is the initial catabolic step of the butanediol oxidation to acetoin.

Reviewer 2 Report

Dear Editor,

The submitted paper is very good in terms of textual structure and scientific content, and it is quite important since it is based on empirical work.

The manuscript shows an interesting topic of the influence of short-chain fatty acids, acetic acid, and dissolved oxygen on 2,3-butanediol production by Saccharomyces cerevisiae W141.  It is quite an interesting approach, and the authors carried out extensive laboratory research.

The suggestions concerning the improvement of these issues along with some typographical and spelling errors:

Line 41 corect ’had’ to ’have’

Line 60 correct ’distrubed’ to ’disturbed’

Line 61 correct ’addition’ to ’ that the addition’

Line 67 correct ’dection’ tp ’detection’

Line 75 Please, consider using ’cultivation’ instead of ’cultural’.

Line 80. The content of yeast cells for inoculation should be provided as the initial cell number, or as yeast dm content.

Line 81, 88 and 89 correct ’ fermentaion’ to ’fermentation’

Line 91. The data on the laboratory vessel type and volume (Erlenmeyer flask or cylinder bottle perhaps) should be provided, along with the way of sealing (perhaps gauze layers, or fermentation sealing cap).

Line 157 Please note that not every reader has access to the previously published works. A short description of metabolite analysis should be provided in this work.

Line 97 insert comma after ’Also’

Line 107 correct ’followed’ to ’ following’

Line 116 Please correct ’internal’ to ’an internal’

Line 125 replace ’were’ with ’was’

Line 126 replace ’8000g’ with ’8000 g’

Line132 correct  ’corporation.’ to ’the corporation.’

Line 137 correct ’ gauze  (4, 8 and 12 layers)’ to ’gauze layers (4, 8 and 12)’

Line 168 correct  particulary’ to ’particularly’; correct ’concentration’ to ’concentrations’

Line 173 The figure is hard to reed. It may be higher resolution or splitted into two figures.

Lines 175-186 There is too much unnecessary information and explanations in the title of the picture, These data are easily visible in the picture. A number of experiments, mean value, and standard deviation are sufficiently explained in section 2. Materials and metods.

Therefore delete ‘From left to right were formic, acetic, propionic, butyric and valeric acid. (A) Bar graph showed the effect of short-chain saturated fatty acids at different concentrations on the yield of 2,3-BDO from S. cerevisiae W141. The vertical coordinates showed the 2,3-BDO yields. (), (), (), (), () and () represented the different concentrations of short-chain fatty acids: 0 g/L, 0.5 g/L, 1.0 g/L, 1.5 g/L, 2.0 g/L and 2.5 g/L, respectively. (B) The point line graph showed the effects of 5 different concentrations of short-chain fatty acids on pH value and the growth of S. 180 cerevisiae W141. Solid points indicated changes in OD600 nm and hollow points indicated changes in  pH. (and), (and), (and), (and) (and), and (and ) represented the different concentrations of short-chain fatty acids: 0 g/L, 0.5 g/L, 1.0 g/L, 1.5 183 g/L, 2.0 g/L and 2.5 g/L, respectively. The data were presented as the means ± standard deviation of three independent experiments (n = 3 each). The error bars represented the standard deviations.’

Please review the titles of all other Figures in the work. They all contain similar unnecessary and too extensive descriptions. Remove the description of SD on every figure title.  

Line 191 correct  particulary’ to ’particularly’;

Line 205 correct ’having’ to ’had’

Line 206 correct ’concentration’ to ’concentrations’

Line 214 correct ’High’ to ’A high’

Line215 correct ’low’ to ’a low’

Line 220 The standard deviation is not visible in the figure. Delete ’The ordinate showed different types of short-chain fatty acids, while the abscissa showed the short-chain fatty acids concentration. The data were presented as the means ± standard deviation of three independent experiments (n = 3 each).

Line 228 Please simplify the title of Figure 3 to ’The effect of pH adjusted by mineral acid (H2SO4) on the (A) 2,3-BDO  production by S.cerevisiae W141, and (B) time course of OD600 nm value of S.cerevisiae W141’

Line 238 delete ’However,’; provide data for optimal concentration

Line 240 correct ’demonstat’ with ’demonstrated’

Line 243-247 Delete unnecessary figure description

Line 262-265 Delete unnecessary figure description

Line 277-278 Delete text.

Line 302 correct ’meant’ to ’mean’
